# Pearl millet genomic vulnerability to climate change in West Africa highlights the need for regional collaboration

Bénédicte Rhoné [1,2,8 ✉], Dimitri Defrance[3], Cécile Berthouly-Salazar[1,4,5], Cédric Mariac[1], Philippe Cubry [1], Marie Couderc[1], Anaïs Dequincey[1], Aichatou Assoumanne[6], Ndjido Ardo Kane [5,7], Benjamin Sultan [3], Adeline Barnaud [1,4,5,9 ✉] & Yves Vigouroux [1,9 ✉]

Climate change is already affecting agro-ecosystems and threatening food security by reducing crop productivity and increasing harvest uncertainty. Mobilizing crop diversity could be an efficient way to mitigate its impact. We test this hypothesis in pearl millet, a nutritious staple cereal cultivated in arid and low-fertility soils in sub-Saharan Africa. We analyze the genomic diversity of 173 landraces collected in West Africa together with an extensive climate dataset composed of metrics of agronomic importance. Mapping the pearl millet genomic vulnerability at the 2050 horizon based on the current genomic-climate relationships, we identify the northern edge of the current areas of cultivation of both early and late flowering varieties as being the most vulnerable to climate change. We predict that the most vulnerable areas will benefit from using landraces that already grow in equivalent climate conditions today. However, such seed-exchange scenarios will require long distance and trans-frontier assisted migrations. Leveraging genetic diversity as a climate mitigation strategy in West Africa will thus require regional collaboration.

[1] DIADE, Univ Montpellier, IRD, Montpellier, France. [2] Univ Lyon 1, CNRS, Laboratoire de Biométrie et Biologie Évolutive UMR 5558, Villeurbanne, France. [3] ESPACE-DEV, Univ Montpellier, IRD, Univ Guyane, Univ Réunion, Univ Antilles, Univ Avignon, 500 rue Jean-François Breton, F-34093 Montpellier Cedex, France. [4] ISRA, LNRPV, Dakar, Senegal. [5] Laboratoire Mixte International LAPSE, Dakar, Senegal. [6] Univ Abdou Moumouni, Niamey, Niger. [7] ISRA, CERAAS, Thiès, Senegal. [8] Present address: AGAP, Univ Montpellier, CIRAD, INRAE, Institut Agro, Montpellier, France. [9] These authors contributed equally: Adeline Barnaud, Yves Vigouroux. ✉email: benedicte.rhone@cirad.fr; adeline.barnaud@ird.fr; yves.vigouroux@ird.fr

Adapting agricultural practices to climate and environmental changes is a major challenge[1–3]. Indeed, climate change is already affecting agricultural productivity[4,5] and future global warming will increase the frequency and intensity of extreme weather such as heat waves and intense rainfall events[6,7] with consequences for food production and hence also for food security[8]. The expected changes in rainfall patterns will affect farming systems dominated by rainfed crops such as in sub-Saharan Africa[9]. Sub-Saharan African agriculture is already impacted and ongoing climate change already reduced yields of major crops by up to 10% up to the beginning of this century[5].

Adaptation strategies have been proposed, from cultivating better adapted varieties or crops to diversify production systems[10–12]. Using existing varietal diversity of crop species and favoring crop varietal replacement is viewed as an efficient short-term strategy to adapt to rapid environmental changes[13,14]. This strategy relies on identifying varieties among currently cultivated, climate-adapted varieties that will still be suitable in future condition. This strategy resembles assisted migration of wild species, which makes it possible to keep pace with climate change in natural ecosystems[15,16].

Ecological niche models are widely used to evaluate the impact of climate change on the distribution of wild species and to provide recommendations for the management of endangered species to minimize future biodiversity loss[17]. Such approaches have most recently been used to identify the parts of the current crop cultivation area that are unlikely to remain suitable for crop production in the future due to climate niche losses[10,18,19]. However, these approaches do not take intra-specific diversity and local adaptation into account when assessing vulnerability. Recent advances that combine genomic diversity with environmental data[20,21] make it possible to mitigate such constraints. These new approaches improve predictions of the impact of future environmental change on crops by accounting for the adaptive potential of the species[22].

In the present study, we use landscape genomic approaches to investigate the impact of climate change on pearl millet cultivation in West Africa. Pearl millet is a staple food for more than 90 million people in the arid and semi-arid tropical regions of Africa and Asia[23,24]. In West Africa, this allogamous cereal is still mainly grown in family farming systems, in rainfed conditions with no additional irrigation, using local varieties (or landraces). High diversity of agro-morphological traits and adaptive traits such as flowering time, photoperiod sensitivity, or drought tolerance exists among and within pearl millet landraces[25–27]. Furthermore, pearl millet is adapted to a wide range of climate conditions with annual precipitation ranging from 200 to more than 1000 mm[26]. Our analyses allow us to identify areas where pearl millet cultivation would be at greater risk under future climate conditions and to assess the agro-biodiversity potential of local varieties to mitigate the impact of climate change. These outcomes suggest that mitigating climate change for traditional African agriculture will require coordinated regional actions and long-scale migration of varieties.

## Results

### Genomic diversity reflects the geographical origin of varieties.
We first built a genomic dataset comprising 173 landraces originating from 10 Sahelian countries (Fig. 1). Using 100 plants per landrace in a pool-sequencing design, we estimated allele frequencies for each landrace at 138,948 polymorphic single-nucleotide polymorphisms (SNPs, Supplementary Data 1 and 2). Principal component analyses (PCAs) of SNP allele frequencies with complete data on landraces clearly separated according to

their geographical origin (Fig. 1c, d) and landraces were found to cluster according to the country.

### Modeling SNP and climate with a GF approach.
We used a gradient forest (GF) approach to model variation in allelic frequencies along environmental gradients[20,21] using both genomic and climate observations at the 173 locations sampled. This approach enables identification of the cut-off in allele frequencies associated with major changes in environmental conditions together with the identification of major climatic drivers of the genomic composition. GF models were built using the 16,632 SNPs with an allele frequency higher than 10%. The climate dataset consisted in 157 metrics of agronomical importance, i.e., onset of the monsoon and other metrics related to precipitation, temperature, and solar radiation (Supplementary Table 1). These metrics were calculated for a period of from 30 to 180 days after monsoon onset, corresponding to the pearl millet growing period, as farmers traditionally sow their fields after the first significant rainfall. These metrics were obtained for 17 climate models (Supplementary Table 2). The climate models were developed by different independent climate modeling groups around the world, associated in the CMIP5 international working group (Coupled Model Intercomparison Project[28]). The 17 models are a subset of the climate models considered by experts of the Intergovernmental panel on Climate Change. GF models were built separately for each of the 17 climate models based on the climate that prevailed on the date the landraces were collected in the field. The most important climate predictors associated with the genomic data are the minimum intensity of solar radiation at the beginning of the growing season, monsoon onset, and precipitation intensity at the beginning of the growing season (Supplementary Table 3). We found an average of 88% of the SNPs that can be predicted by the environmental variables (from 14,544 to 14,757 SNPs with a $R^2 > 0$) depending on the climate model considered (Supplementary Table 3).

### Genomic vulnerability shows a latitudinal pattern.
Based on the current climate–genome relationship modeled using this GF approach, we then predicted the genomic composition expected in the future. The future genomic composition was predicted throughout the pearl millet cultivation area in West Africa at the 2050 and 2100 horizons from future climate projections made by a dedicated climate model. We then computed the genomic vulnerability as the Euclidean distance between current and future genomic compositions. Genomic vulnerability measures the distance between the genotype of the currently cultivated landraces and the inferred genotype under the future climate using modeled genotype/climate relationships. Thus, genomic vulnerability is a measure of how much genetic change is needed to adapt to ongoing climate changes. Genomic vulnerability relates to the risk of non-adaptation or to the evolutionary change required to cope with the future environment. To account for variability between climate models, we predicted the genomic vulnerability of each of the 17 climate models (Supplementary Table 2). For the future climate projections, we also considered two different scenarios of a representative concentration pathway (RCP[29]) of greenhouse gas emission trajectories among the four adopted by the CMIP5 consortium as follows: (i) RCP2.6, a scenario based on the assumption of a strict reduction in carbon dioxide over the years that will limit the increase in global temperature to <2 °C by 2100; (ii) RCP8.5, a pessimistic scenario based on the assumption that emissions will continue to increase leading to a 4 °C increase in temperature by 2100. Genomic vulnerability inferred as the mean across the 17 models revealed a similar spatial pattern under the two RCP scenarios considered at

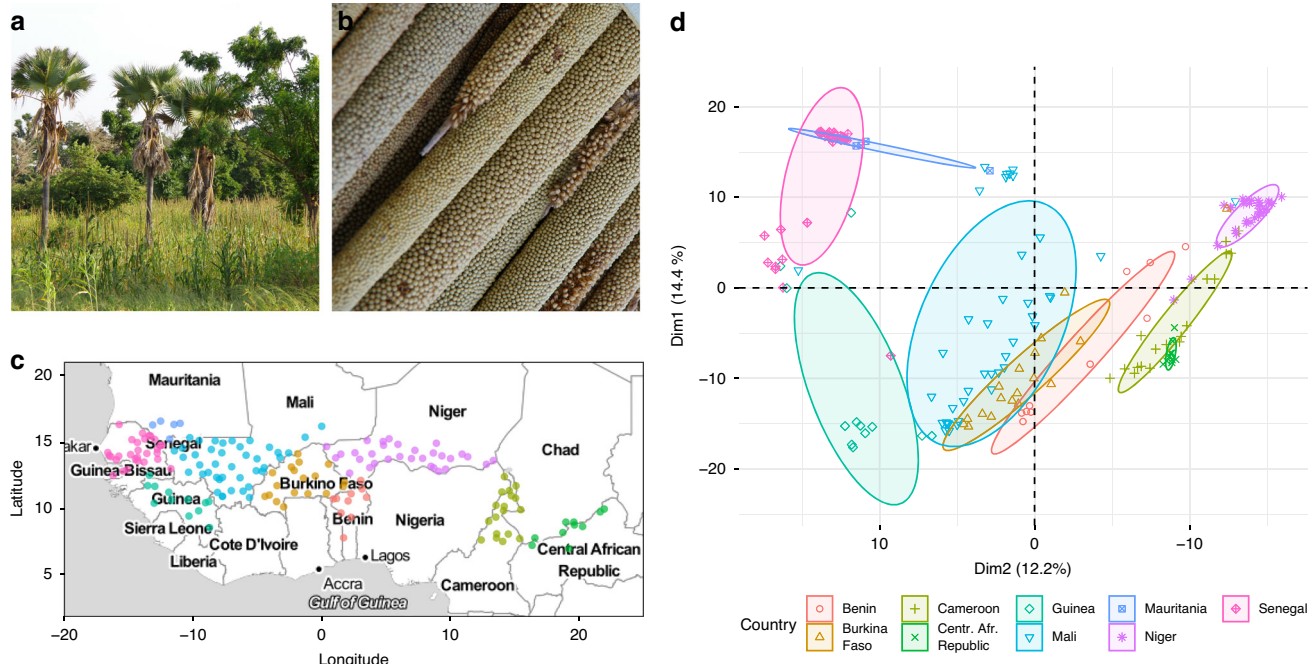

**Fig. 1 Diversity of pearl millet varieties in West Africa. a** Photo of a pearl millet field in Senegal ©A. Barnaud, IRD. **b** Early-flowering pearl millet panicle from Senegal, © C. Berthouly-Salazar, IRD. **c** Map of the 173 accessions of pearl millet considered in this study. **d** Principal component analysis (PCA) of pearl millet landraces based on SNP allele frequencies. The plot shows landrace PCA projections on the first two axes that explain 26.6% of total variance. Landraces are colored according to the country. Concentration ellipses with a normal probability of 0.6 are drawn around landraces originating from the same country.

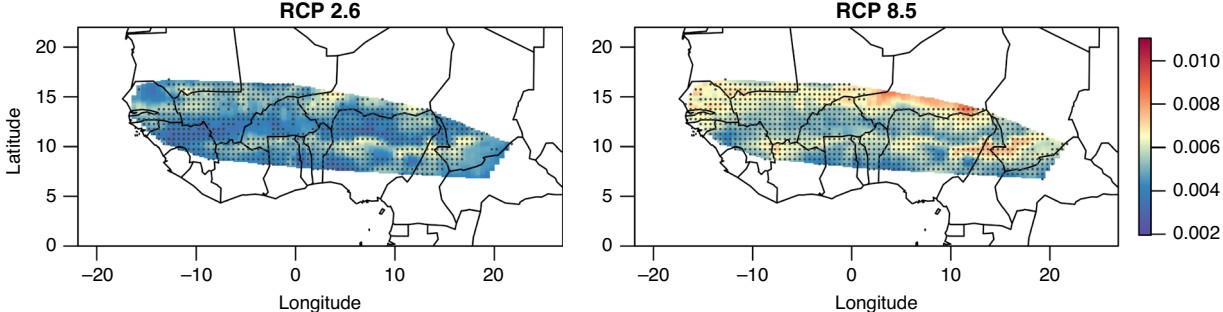

**Fig. 2 Pearl millet genomic vulnerability to climate change at the 2050 horizon.** Genomic vulnerabilities in West Africa were estimated based on projections for two scenarios of gas concentration pathways RCP2.6 and RCP8.5. The color scale refers to the mean value of genomic vulnerability estimated for 17 climate model projections. Stippling identifies areas where the magnitude of the mean genomic vulnerability is more than twice the SD (i.e., coefficient of variation <50%), identifying regions where mean genomic vulnerability estimated using multi-model climate projections is high compared to the inter-model estimates of variability.

the 2050 horizon, but that varied in intensity (Fig. 2 and Supplementary Fig. 1). The use of all climatic variables or of only uncorrelated climatic variables had no major impact on our results (Supplementary Fig. 2). Genomic vulnerability displayed latitudinal organization with higher vulnerability around latitude 10° and latitude 15°. It should be noted that some areas, e.g., Niger in the RCP8.5 scenario, were associated with a high coefficient of variation across models (CV > 50%), indicating contrasting patterns of vulnerability depending on the climate model used (Fig. 2 and Supplementary Fig. 1). This is probably linked to the high uncertainty of climate model projections of changes in precipitation in such regions[30].

**Genomic vulnerability is associated with flowering time.** The latitudinal pattern of genomic vulnerability we found in West Africa (Fig. 3a, b) may be associated with the length of the pearl

millet flowering period. Mean flowering time in our 173 land-races ranges from 41 to 124 days (Supplementary Fig. 3). Flowering time exhibited bi-modal distribution, with a first mode of around 60 days in the early-flowering landraces and a second mode of around 110 days in the late flowering landraces. The distribution of flowering time is spatially organized along a latitudinal gradient, long-cycle landraces being cultivated in the south up to 11° latitude and short-cycle varieties being cultivated in the north from latitude 13° to 16° (Fig. 3c). Crossing the distribution of genomic vulnerability with the length of flowering time revealed greater vulnerability at the northern limits of cultivation in both early and late flowering landraces (Fig. 3d). By contrast, reduced vulnerability was predicted for late flowering types in their southern cultivation area or at latitudes where a mix of both types plus types with intermediate flowering periods are cultivated.

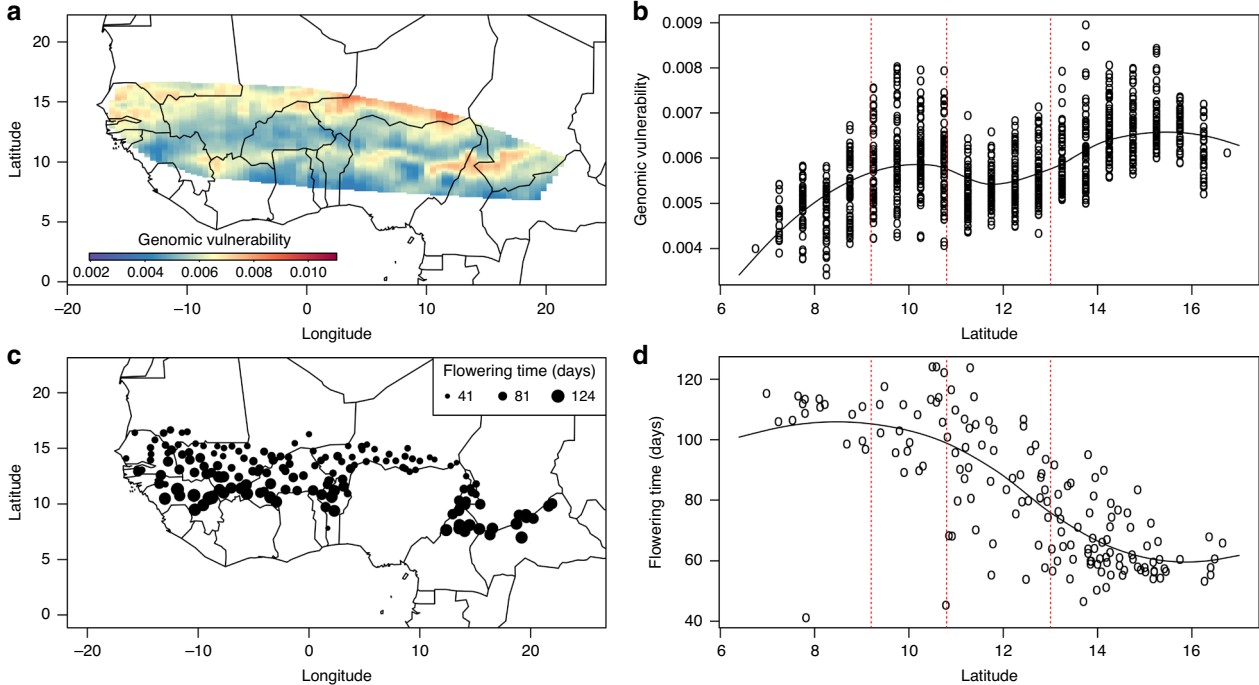

**Fig. 3 Spatial distribution of genomic vulnerability relative to flowering time. a** Genomic vulnerability of pearl millet to climate change at the 2050 horizon. The gas emission scenario considered here is RCP8.5. **b** Genomic vulnerability of pearl millet cultivation as a function of latitude. Circles indicate the genomic vulnerability within the cultivation area plotted as a function of the latitude concerned. The curve of genomic vulnerability with respect to latitude was smoothed using the locfit R function. Red dotted lines separate latitudinal ranges with higher genomic vulnerability. **c** Flowering time of landraces depending on their location of origin. The size of the circle indicates the average flowering time. **d** Flowering time of the landraces as a function of latitude. The curve of flowering time as a function of latitude was drawn using the locfit R function. Red dotted lines separate latitudinal ranges with the highest genomic vulnerability.

To validate the hypothesis that flowering time is an important trait in adaptation to climate, we first performed a genome-wide association study (GWAS) to find SNPs linked to flowering time. Association analysis was performed using a latent factor approach (latent factor mixed model[31,32], LFMM). We considered five confounding factors to control for population structure ($K = 5$). Using more confounding factors, i.e., up to seven, did not lead to better control of population structure in the model and did not alter the final results (Supplementary Figs. 4–6). A total of 103 SNPs were found to be significantly associated with flowering time ($K = 5$, Supplementary Fig. 5). These SNPs are mainly found on chromosomes 1, 2, and 5 on the pearl millet genome and correspond to 75 annotated genes (Supplementary Data 3).

We then investigated whether the SNPs associated with flowering time were also better predicted by the climate variables in the GF model than the other SNPs. To this end, we used the proportion of variance ($R^2$) explained by the climate predictors for each SNP. The 103 SNPs associated with flowering time presented an average correlation twice higher (mean($R^2$) = 0.53) than all the SNPs considered (mean($R^2$) = 0.28, Wilcoxon rank test, $p$-value < $2.10^{-16}$, Supplementary Fig. 6). Consequently, SNPs associated with flowering time contribute strongly to the prediction of genomic vulnerabilities to future climate. This result reinforced our previous observation that flowering time is a major trait of pearl millet adaptation to climate.

**Genomic vulnerability is associated with yield**. We expect the genomic vulnerability statistic to be negatively correlated with yield. Greater genomic vulnerability should be associated with yield loss. As assessing yield under future climate might be difficult, we used spatial contrast of climate to experimentally link genomic vulnerability with yield. To this end, we used the two-year common garden experiment on pearl millet landraces conducted in Sadoré (Niger). In this case, the genomic vulnerability of a given landrace was predicted using the climate condition at the location of origin of the landrace and the climate at the Sadoré experimental site. We found significant negative correlations between yield-related traits and genomic vulnerability ($r$(Pearson) = −0.412, $p < 0.0001$ for 100-seed weight; $r$(Pearson) = −0.368, $p < 0.0001$ for the mean weight of seeds on the main spike; $r$(Pearson) = −0.310, $p < 0.0001$ for the total weight of seeds per plant, Supplementary Figs. 7 and 8). These results indicate that higher genomic vulnerability is associated with lower fitness of the landraces under the climatic conditions at the site of the field experiment. The correlation with yield-related traits was slightly lower when genomic vulnerability was estimated using the subset of uncorrelated climate metrics ($r$(Pearson) = −0.374, $p < 0.0001$ for 100-seed weight; $r$(Pearson) = −0.252, $p = 0.0015$ for the mean weight of seeds on the main spike; $r$(Pearson) = −0.258, $p = 0.0011$ for the weight of seeds per plant). We consequently decided to only keep the genomic vulnerability assessed using all climate metrics in the following analyses.

**Inter-country migration could mitigate the impact of future climate**. To infer how migration could help reduce the impact of climate change on yield, we first determined the most vulnerable regions identified by each of the 17 climate models. Two to eight vulnerable areas were identified per climate model at the 2050 horizon under the RCP8.5 scenario. A total of 80 vulnerable areas were identified considering all the climate models together (Fig. 4a). We then assessed the distance and origin of the current landrace that could be used to mitigate the impact of future

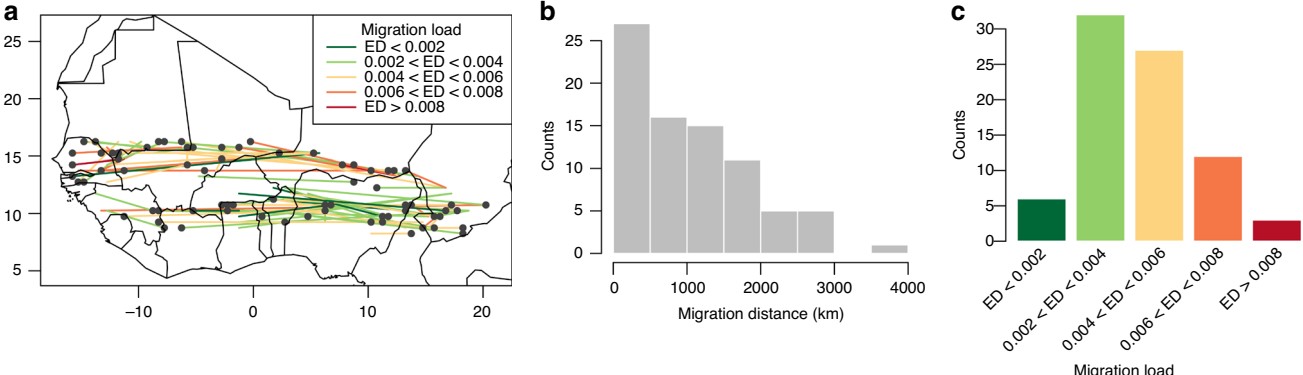

**Fig. 4 Assisted migration of pearl millet varieties for adaptation to future climate at the 2050 horizon.** All the figures are based on the RCP8.5 scenario. **a** Migration trajectories are indicated by colored lines linking one of the 80 most vulnerable locations (black dots) to the location selected for optimal migration of varieties to minimize future genomic vulnerability. **b** Migration distance distribution. **c** Migration load distribution. The migration load refers to the genomic vulnerability of landraces after migration under future climate projections. High migration load values identify migrations that rely on potentially ill-adapted migrated varieties as no other varieties that are better adapted to future climate conditions exist in the cultivation area.

climate change in a given vulnerable region (Supplementary Fig. 9). We selected the landrace to migrate by choosing the one with the lowest genomic vulnerability to future climate conditions in the vulnerable region. We called this optimal migration. The optimal migration distances ranged from 77 to 3665 km with a mean distance of 1059 km (SD: 801 km, Fig. 4b). A total of 88.3% migrations would be between countries. We used the genomic vulnerability of the migrated landrace to the future climate condition projected at the given vulnerable region as a measure of migration load. This measure allowed us to reveal where the migrated landrace would be not perfectly adapted. High migration load values indicate migrations that rely on potentially ill-adapted migrated varieties as no other varieties that are better adapted to future climate conditions exist in the cultivation area. The migration load ranged from 0.0010 to 0.0215 (mean = 0.0045; Fig. 4c). The high vulnerability of some migrated landraces (migration load >0.01) suggests that some climate model projections for the 2050 horizon do not correspond to the climate faced by varieties cultivated today. Consequently, the proposed migrations could rely on landraces that are ill suited for the purpose.

We investigated whether less optimal migration scenarios could reduce the migration distance. We selected the closest landrace to migrate among the 1% lowest genomic vulnerability (near-optimal migration) or among the 5% lowest genomic vulnerabilities (sub-optimal migration Supplementary Figs. 10–12). For near-optimal migration, the mean migration distance is 537 km with 63.75% of transboundary migration and a mean migration load of 0.0052 (range 0.0014–0.0215, Supplementary Fig. 7). For sub-optimal migration, the mean migration distance is 257 km with 37.5% of transboundary migration and a mean migration load of 0.0062 (range 0.0024–0.0215). Thus, using less optimal migration scenarios reduces the migration distance but also increases the migration load by 10–30%.

## Discussion

Adaptation of today's agriculture to climate change requires the assessment of crop vulnerability[18]. In this study, we combined spatial genetic structure and spatial climate variability[21,33] to assess crop vulnerability to future climate change and how assisted migration could help mitigate its effect. By integrating species genetic diversity in the predictive model, this approach expanded existing methods that only focus on plant distribution,

e.g., in ecological niche modeling[19]. Our approach makes it possible to account for local adaptation nested in the population structure but could also directly rely on alleles linked to local adaptation. With the increasing availability of crop genomic resources, we should be able to predict which combination of adaptive alleles will be suitable in given future conditions. Another breakthrough is the use of multi-model projections of relevant climate metrics for the pearl millet growing period aligned with the monsoon onset. Using a common garden experiment, we show that genomic vulnerability assessed using our approach is associated with yield-related traits, varieties with higher vulnerabilities exhibiting the biggest yield loss. This experiment thus validates the predictive potential of genomic vulnerability and its biological relevance for agriculture by statistically linking this estimate with the yield.

The northern edge of the cultivation areas of both early and late flowering pearl millet was found to be the regions with the highest genomic vulnerabilities. Flowering time is an important trait for yield elaboration in annual plant species, as it synchronizes reproduction with climate conditions, thereby ensuring grain filling under favorable environmental conditions[34–36]. An increase in temperature combined with a reduction in precipitation is likely to affect late flowering varieties, in particular by increasing evapo-transpiration at maturation[5]. Our results thus underline the importance of taking flowering time into consideration when studying climate adaptation in pearl millet.

Crop diversity[11] and varietal diversity[37] will make it possible to mitigate the impact of climate variation. Pearl millet landraces are characterized by highly diverse agro-morphological and adaptive traits, including flowering time, photoperiod sensitivity, or drought tolerance[25–27,38]. Building on existing diversity, we investigated whether a varietal replacement strategy involving migration of better adapted varieties to future climate conditions could mitigate yield loss and reduce agricultural risks in vulnerable areas. Our analyses show that a strategy involving long distance and mainly transboundary migration will indeed be able to partly mitigate ongoing climate change. These results highlight the need for the discussion of action plans at the scale of West Africa as a whole. The need for regional collaboration in West Africa for climate mitigation has already been previously acknowledged[39] based on present and projected future climate analogy.

In family farming systems, farmers are directly involved in crop varietal innovation and in the management of crop genetic

resources[40]. Varietal diffusion is shaped by social factors, as illustrated by the relationship between the structure of crop diversity and ethno-linguistic groups[41–43]. Farmers' seed networks are known to be major drivers of gene flow, within and beyond local communities and environments[44,45]. In some cases, this traditional seed system may be able to provide farmers with landraces suitable for agro-ecological conditions under predicted climate change, as already demonstrated for maize landraces in Mexico[46]. However, our findings also clearly show that farmers are likely to need to source seeds outside their traditional geographic ranges, to be sure germplasm is suitable for cultivation in the future[46].

With the exception of strategies that involve simple changes in agricultural practices, such as crop planting schedules, most adaptation strategies to mitigate the consequences of climate change on agriculture require significant investment by farmers and substantial modifications of eating behavior and social organization[3]. This also applies to the relocation of crop production to more favorable climatic regions. This kind of adaptation strategy either requires farmers to migrate or crop species to be replaced by species that are better adapted to the new environmental conditions[10], and consequently leads to significant changes in the eating habits of the population concerned. One key factor that needs to be taken into consideration when exploring varietal replacement strategies is farmers' potential resistance to the adoption of migrated varieties. Involving farmers in participatory breeding schemes could help solve this problem, while making it possible to better meet farmer's needs for varietal adaptation to climate change[47–49].

## Methods

**Plant material**. A total of 173 geo-referenced landraces originating from 10 West African and Central African countries were sampled. The landraces were collected by the Institut de la Recherche pour le Développement (IRD) on dedicated missions conducted between 1974 and 1989 (Supplementary Data 1). One hundred seedlings of each of the 173 landraces were grown in IRD greenhouse facilities in Montpellier (France) from seed stocks maintained as part of the IRD collection. This corresponds to a total of 17,300 seedlings grown for DNA extraction. The same quantity of leaf material was collected from each of the 100 seedlings of each landrace for DNA extraction.

**Pool-sequencing, bioinformatic analysis**. *Bait design*. A total of 152,619 biotinylated 80 bp baits were designed and synthesized by Mycroarray (Ann Arbor, Michigan, USA, reference: 160606-32). The baits target the first 1000 bp of all annotated genes of the genome. Repetitive sequences over the pearl millet reference genome were discarded using RepeatMasker (http://www.repeatmasker.org).

*Library preparation and sequencing*. Libraries were prepared according to the protocol detailed in ref. [50]. Briefly, DNA samples were sheared to yield 400 bp fragments. DNA was then repaired and tagged using 6 bp barcodes to allow further multiplexing. Real-time PCR was performed to complete adapter sequences and to generate ready-to-load libraries. The libraries were either immediately sequenced for shotgun genomic sequencing or enriched by capture using the Myselect protocol (Mycroarray) before sequencing. Sequencing was performed using four Illumina sequencing lanes on a HiSeq2500 and outsourced to Novogen in China.

*Read filtering and mapping step*. Adapter sequences and extremities with low quality scores (–q 20) were removed from row reads using Cutadapt (v1.10[51]). Reads were filtered based on their length ($L$ > 35 bases) and on their quality mean values ($Q$ > 30) using a freely available Perl script (https://github.com/SouthGreenPlatform/arcad-hts/blob/master/scripts/arcad_hts_2_Filter_Fastq_On_Mean_Quality.pl). The separately trimmed forward and reverse reads were then re-synchronized into pairs with an in-house Perl script. The filtered paired-end reads were mapped to the pearl millet reference genome[14] using BWA (Burrows-Wheeler Aligner, v0.7.2[52]), with the MEM algorithm and standard parameters. Unmapped, low mapping quality (MAPQ < 20) and improperly paired reads were filtered out with SAMtools (v1.1[53]). The exome mean coverage was estimated for each accession with QualiMap (v2.2[54]) using the gene annotation from ref. [14]. Local realignment was performed using IndelRealigner and variant calling was performed using UnifiedGenotyper in the Genome Analysis Toolkit (GATK v3.7[55]).

*SNP calling*. Raw SNPs were filtered out using the GATK VariantFiltration tool with the following criteria: bi-allelic SNP, depth coverage >10 and <250 per accession, <3 SNPs in a window of 5 bp, frequency of the alternate allele to be called as an SNP (AF > 0.003 corresponding to a minimum count of five reads with the alternative allele throughout the dataset). For each accession, SNPs with a total read count of <20 reads were set to NA. Finally, only the SNPs with complete data were considered in the final SNP set. The allele frequencies of the remaining SNPs after filtration were estimated and total read counts using an in-house R script. SNP filtration criteria were chosen to maximize the correlation of allele frequencies obtained from three PE05487 sequencing replications. Accession PE05487 was sequenced twice pooled (i.e., two sequencing runs from the same library). In addition, this accession was sequenced from 100 distinct unpooled individuals. The unpooled individual fastq files were submitted to the very same bioinformatic pipeline as described below and allelic frequencies were obtained from the.vcf file generated by UG-GATK after variant filtration. Our bioinformatic pipeline resulted in a highly correlated estimation of allele frequencies at common SNPs ($r$(Pearson) = 0.91 to 0.96, $n$ = 5 275).

*PCA analysis*. PCA analysis was performed on allelic frequencies using the prcomp() R function to summarize variations in population structure.

**Climate data**. *Climate datasets*. The data in the observed climate dataset were extracted from the EWEMBI climate dataset[56]. The EWEMBI dataset covers the entire globe at 0.5° horizontal and daily temporal resolution from 1979 to 2013 and combines several reference datasets: ERA-Interim reanalysis data[57], WATCH forcing data methodology applied to ERA-Interim reanalysis data[58], eartH2Observe forcing data (E2OBS[59]), and NASA/GEWEX Surface Radiation Budget data (SRB[60]). The SRB data were used to bias-correct E2OBS shortwave and longwave radiation[61]. Variables included in the EWEMBI dataset are Near Surface Relative Humidity, Near Surface Specific Humidity, Precipitation, Snowfall Flux, Surface Air Pressure, Surface Downwelling Longwave Radiation, Surface Downwelling Shortwave Radiation, Near Surface Wind Speed, Near Surface Air Temperature, Daily Maximum Near Surface Air Temperature, Daily Minimum Near Surface Air Temperature, Eastward Near Surface Wind, and Northward Near Surface Wind. A sample of this dataset was extracted for a period corresponding to the sample collecting missions among the available data in the dataset (1979–1989, mean value over the 10-year period) at a pixel resolution of 0.5° × 0.5° (~50 × 50 km). In the present study, the pearl millet cultivation area is delimited by the convex hull of our landrace coordinates. This area extends over 3.1 million km$^2$, including 1041 pixels. Climate data for the sampled locations were inferred from the pixel in which the landrace coordinates are located.

Future climate projections consisted in bias-corrected daily climate data extracted from a subset of 17 climate models among the CMIP5 dataset[28] (Supplementary Table 4). Climate data from the 17 climate models were corrected using the Cumulative Distribution Function transfer method to reduce errors in the present-day simulations of the CMIP5 models compared to observed data[28,58,62,63]. We focused our analysis on two RCPs (RCP2.6 and RCP8.5), representing two future greenhouse gas concentration trajectories[29]. Only the 17 climate models that include both the RCP2.6 and the RCP8.5 scenarios were used. For each climate model and RCP scenario, we extracted climate projections for the 2049–2059 period, hereafter referred to as T2050, and for the 2089-2099 time period, hereafter referred to as T2100. The so-called historical climate projections for each of the 17 CMIP5 models were also extracted for the 1979–1989 period for vulnerability assessment.

*Climate metrics*. Climate observations and future climate projections were used to compute 157 metrics in five categories (Precipitation, Mean Near Surface Air Temperature, Near Surface Maximum Air Temperature, Near Surface Minimum Air Temperature, and Surface Downwelling Shortwave Radiation; Supplementary Table 1), which are critical variables for agricultural purposes. The monsoon onset was assessed from rainfall data in each pixel in the cultivation area and for each model. This metric has been found to be highly correlated with the sowing date of pearl millet in the West African cultivation area[64]. All the other 156 climate metrics were mean, minimum, or maximum values of a climate parameter or number of events calculated within a period of 30, 60, 90, 120, 150, or 180 days after monsoon onset (6 × 26 metrics).

**GF predictions of genomic vulnerability**. We used GF (R package, gradientForest[21,33]) to model the importance of changes in allele frequency along the environmental gradients. GF is a machine-learning approach derived from the random forest algorithm. It is based on regression trees linking allele frequencies with environmental data observed at the locations of the 173 landraces. For this analysis, 500 trees per SNP were generated. We only considered SNPs with a minor allele frequency >10% following[20]. GF analysis enabled identification of a list of climate predictors ranked in order of importance and the SNPs with predictive power (i.e., with $R^2$ > 0). This $R^2$ measures the proportion of variance explained by the climate predictors for each SNP calculated using a cross-validation procedure (see refs. [21,33] for details). Using the turnover functions generated by GF analysis linking allele frequency changes with environmental variables, we predicted the genomic composition expected at unsampled locations or, for the future, based on climate data projections. Current and future genomic composition throughout the pearl millet cultivation area in West Africa were thus predicted using the historical and future predictions in each of the 17 climate models. As in ref. [20], genomic vulnerability (also referred to as genomic offset in ref. [21]) was calculated as the Euclidean distance between the genomic composition under the past and future projected climates. This analysis enabled assessment of the genomic vulnerability of

pearl millet landraces currently cultivated in West Africa at the 2050 and 2100 horizons. Genomic vulnerability was obtained using both the dataset of total climate metrics and the subset of uncorrelated climate metrics following[20] with a maximum Pearson's correlation threshold of 0.7.

**GF model assisted migration scenarios**. We wanted to find out which currently cultivated landraces would be best adapted to future climate conditions projected in the most vulnerable areas in order to infer migration. This analysis relied on the GF models previously built for each of the 17 climate models. Each climate model was analyzed separately and then combined.

For a given climate model, we identified vulnerable areas using a spatial approach that clustered the 10% most vulnerable pixels (see details in Supplementary Fig. 9). Clusters of vulnerable pixels were designed using DBscan (Density-Based Spatial Clustering of Applications with Noise; dbscan R package[65]), which groups pixels that are closely packed together. Clustering was based on the geographic distance between vulnerable pixels measured with the distm function in the geosphere R package. Only clusters with at least four vulnerable pixels and separated by <1200 km were retained for further analysis. This distance corresponds to the value that best optimizes the spatial distribution of the vulnerable areas within the 3.1 million km$^2$ region considered here. To assess migration, in each cluster, we selected the pixel with the highest vulnerability.

For each cluster independently (Supplementary Fig. 9C, D), a genomic vulnerability estimate (i.e., Euclidean distance) was calculated between the future climate predicted in the cluster and current climate conditions in other pixels covering the entire cultivation area. The lowest genomic vulnerability estimate (i.e., the minimum Euclidean distance, EDmin) pinpoints the location where the landrace best adapted to the future climate condition of the highly vulnerable area might be found. In this way, we identified the location of the landrace that is currently the best adapted to the future climate in the vulnerable area for optimal migration of varieties. We then measured the geographic distance between the identified location and the vulnerable area and used the genomic vulnerability estimate (EDmin) as a measure of migration load. This measure represents the genomic gap that needs to be filled for the migrated varieties to be fully adapted to their new location and conditions. High migration loads indicate migrations relying on migrated varieties that may not be perfectly adapted, as no other varieties that are better adapted to future climate conditions in the vulnerable area exist in the cultivation area.

We also checked whether migration distance could be reduced by selecting the pixel located closest to vulnerable area among the 1% least vulnerable pixels for a near-optimal migration instead of the optimal migration (Supplementary Fig. 9). We also investigated one sub-optimal migration corresponding to the 5% of least vulnerable pixels. Migration distance and migration load were thus assessed under optimal (EDmin), near-optimal (Closest_EDmin1%), and sub-optimal (Closest_EDmin 5%) conditions.

**Measuring flowering time**. Flowering time was measured in six trials composed of two completely randomized blocks in 2016 and four trials in 2017. All the trials were performed at the International Crops Research Institute for the Semi-Arid Tropics field station in Sadoré, Niger (Lat. 13.2375, Long. 2.2797). Ten plants were phenotyped for each landrace. Sowing dates ranged from 15 June to 6 July, depending on the trial. The trials were conducted under rainfed conditions with supplementary sprinkler irrigation if necessary.

**GWAS analysis for flowering time**. Analyses associating SNP allele frequencies with flowering time were performed using a subset of 27,409 SNPs with a minor allele frequency >5%. We adjusted both a simple linear model that did not account for population structure and a LFMM (with the lfmm R package[31,32]) designed for the correction of unobserved confounding factors such as population structure. We estimated the number of latent factors from the screeplot generated from a PCA of the genomic data with the prcomp R function. After model selection based on $Q–Q$ plots (Supplementary Fig. 4), we considered association results obtained from the LFMM analyses with five latent factors. We used a false discovery rate (FDR) of 5% to select associated SNPs (with the qvalue R package, v 2.18.0). Intersecting the position of the SNPs with the position of the pearl millet genes[24] using the bedtools program (v2.27.1), we identified annotated candidate genes involved in variations in flowering time.

The GF model linking change in allele frequencies with climate provides a correlation ($R^2$) for each SNP by assessing their relative importance in the model based on the cross-validation procedure detailed in ref. [33]. We compared the mean $R^2$ of the SNPs associated with flowering with the mean $R^2$ of all the SNPs using a Wilcoxon rank test. The $R^2$ values of each SNP were obtained from the GF model built with observed current climate data (EWEMBI dataset).

**Analysis of genomic vulnerability and yield**. As genomic vulnerability measures the mismatch between current and predicted future genomic variation[20], the biggest mismatches, i.e., vulnerabilities, should result in maladaptation and hence in less fit varieties. A negative relationship between yield (as a proxy of fitness) and genomic vulnerability could therefore be expected. We experimentally assessed the

relationship between yield and genomic vulnerability using the spatial heterogeneity of climate experienced by pearl millet varieties in West Africa. For that purpose, we first evaluated the yield of landraces originating from West Africa under the common garden conditions at Sadoré, Niger (Supplementary Fig. 7). We used yield-related measurements based on the main spike in 10 plants of each landrace sowed. The yield-related measurements were 100-seed weight, the mean weight of the seeds on the main spike, and the average weight of seeds per plant. The last was estimated by multiplying the mean weight of seeds on the main spike and the mean number of productive tillers in each variety. We then calculated the genomic vulnerability of these landraces under the conditions used in the field experiments, measured here as the Euclidean distance between the genomic composition predicted under the existing climate conditions in the location of origin of the landraces and the climate of the field experiment (Sadoré). Genomic vulnerability was assessed using the GF model built from the observed climate data (EWEMBI dataset).

**Reporting summary**. Further information on research design is available in the Nature Research Reporting Summary linked to this article.

## Data availability
The raw sequencing data are deposited in the NCBI Sequence Read Archive (SRA) database with the BioProject accession number PRJNA422966. The allele frequency data, the raw phenotypic data, and the climate datasets associated with this paper are publicly available in the Zenodo repository at https://doi.org/10.5281/zenodo.3970815[66].

## Code availability
The scripts for the bioinformatics analysis and the custom R code used to perform the analyses are publicly available in the Zenodo repository at https://doi.org/10.5281/zenodo.3970815[66].

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

## Acknowledgements

The research leading to these results received funding from the UK's National Environment Research Council (NERC)/Department for International Development (DFID) Future Climate For Africa program, under the AMMA-2050 project (grant numbers NE/M020002/1 and NE/M019934/1) and from Agropolis Fondation under AdaptInCrops project (ID 1403-057). Y.V. is also funded by the CGIAR Research Program on Grain Legumes and Dryland Cereals (GLDC). We acknowledge the IRD itrop HPC (South Green Platform) at IRD Montpellier for providing HPC resources that have contributed to the research results reported within this paper, URL: https://bioinfo.ird.fr/- http://www.southgreen.fr. We thank Ndjibo Moussa and Tidjani Moussa for field trial analyses.

## Author contributions

A.B. and Y.V. jointly managed the study. B.R., A.B., C.B.-S., N.K., A.A., B.S., and Y.V. designed the study. B.R. performed the analysis. C.M., M.C., and A.D. generated the genomic datasets. Y.V. and A.A. managed the field studies. B.S. and D.D. provided and analyzed the climate data. P.C. helped with GWAS analyses. B.R. wrote the paper in collaboration with A.B. and Y.V., with input from all authors.

## Competing interests

The authors declare no competing interests.
