## [Peer Review File · Nature Communications]

Reviewers' Comments:

Reviewer #1:

Remarks to the Author:

This is an interesting paper that attempts to examine the landscape distribution of pearl millet in West Africa, the climatic associations of the distribution. And projected optimal growing areas under different climate change scenarios. The authors find that flowering time may be a key trait associated with geography/climate and that long-range migration of current genotypes maybe necessary for continued yield under various climate change scenarios.

While this is an interesting and timely paper, there seemed to be a lack of detail and depth in the manuscript. Here are some issues:

1. It would be good to define genomic vulnerability – what is it? It appears to be defined in terms of a Euclidean distance (that should be placed in the main text), but what does it mean biologically?
2. There seemed to be a lack of description in the results. The authors gave broad patterns and conclusions, but it feels like more detail is needed to thoroughly explore their results in the main text.
3. To be suitable for the current journal, it would be very helpful if more detailed genetic analysis was conducted. Can you map genome vulnerability? How about flowering time, which they recognize as a key trait - surely this trait and others can be analyzed by GWAS?
4. There also was very little in the text on the climate scenarios – what are the 2.6 and 8.5 scenarios, how likely are they, what are the consequences across the landscape, etc.

Reviewer #2:

Remarks to the Author:

In "Pearl millet genomic vulnerability to climate change in West Africa highlights the need for regional collaboration", Rhoné et al investigated if the genetic diversity of pearl millet can be used to mitigate adverse effects of climate change. The authors found that the most vulnerable areas in the future would benefit from seed exchange, especially from those regions where today's climate is similar. Even though the conclusion might be obvious, showing where current landraces will have value in the future is of great interest, especially for stakeholders. The study and manuscript are well constructed, and the results thoroughly discussed. I particularly enjoyed the application of the gradient forest algorithm.

In fact, I have only picked up a few things to correct and suggest that the manuscript is accepted with minor revisions.

L25: "cultivated plants" -> crops?

L33: I suggest: "leveraging genetic diversity as a climate mitigation strategy in West Africa will..."

L71: SNPs has not yet been defined.

L128: define migration load here not in the figure caption.

Point by point response to the reviewers

Object: NCOMMS-20-17234

Dear reviewers,

We appreciated the positive reception of the article by the two reviewers.

We proposed a revised version of our manuscript NSCOMMS-20-17234 entitled “Pearl millet genomic vulnerability to climate change in West Africa highlights the need for regional collaboration” by Rhoné et al.

For this resubmission, we provide a substantially revised manuscript, respecting the main text limit of 5,000 words. This major revision of the manuscript includes text revision for clarity and level of details and the additional GWAS analysis for flowering time recommended by reviewer #1 and yourself. As Phillipe Cubry was associated to these additional analyses, we added him as a co-author of the revised manuscript.

Additional text in the manuscript text file is highlighted in the text file with color highlighting.

We address all the specific reviewer requests and provide a detailed response to reviewers in the point-by-point response in the following part of this letter.

Sincerely yours,

Point-by-point response to the reviewers' comments

Response to reviewer #1

“While this is an interesting and timely paper, there seemed to be a lack of detail and depth in the manuscript.”

Our response:

We substantially enriched the text, adding more explanation and details on the analyses, in particular in the results section (for instance L94-102: we add more details on the climate models, L115-124: we provide more explanation on the genomic vulnerability measure and significance). We thus provide here a main text version increased by 1100 words compared to the previous version of the manuscript. We think the quality of the paper is now improved.

“1. It would be good to define genomic vulnerability – what is it? It appears to be defined in terms of a Euclidean distance (that should be placed in the main text), but what does it mean biologically?”

Our response:

In this revised version of the manuscript, we provide a definition of the concept of genomic vulnerability in the main text and not only the material and methods section. (L115-119: *“Based on the current climate-genome relationship previously modeled with the gradient forest approach, we predicted the genomic composition expected in the future. The future genomic composition was predicted throughout the pearl millet cultivation area in West Africa by 2050 and 2100 from future climate projections of a given climate model. We then computed the genomic vulnerability as the Euclidean distance between current and future genomic compositions.”*).

We also precise the biological meaning of the concept in the main text (L120-124: *“This parameter measures the distance between the genotype of the current cultivated landraces and the inferred genotype under the future climate using modelled genotype/climate relationships. Thus, genomic vulnerability gives a measure of how much genetic change is needed to adapt to ongoing climate changes. Genomic vulnerability relates to the risk of non-adaptation or to the evolutive step needed to cope with future environment.”*)

We also show that genomic vulnerability is associated with fitness of the landraces using a common garden experiment, linking directly this parameter to yield loss (L171-187).

“2. There seemed to be a lack of description in the results. The authors gave broad patterns and conclusions, but it feels like more detail is needed to thoroughly explore their results in the main text.”

Our response:

We add more description on the performed analysis and comprehension details over all the results section, thus doubling the number of words in the section compared to the previous version.

“3. To be suitable for the current journal, it would be very helpful if more detailed genetic analysis was conducted. Can you map genome vulnerability? How about flowering time, which they recognize as a key trait - surely this trait and others can be analyzed by GWAS?”

Our response:

Genomic vulnerability is calculated as a distance. Thus, it is difficult to directly map it to the genome. However, as you suggest, we analyzed if SNPs significantly associated with an adaptive phenotype (flowering time) contribute strongly to its estimation.

We performed the suggested GWAS analysis for flowering time and add the new result in a new paragraph (L154-161). The analysis nicely shows that the SNPs associated with flowering time have a high weight in the gradient forest model used to calculate genomic vulnerabilities (L161-169). So yes,

genomic vulnerabilities is built upon functional variants associated with adaptation as illustrate with this analysis. We think this additional analysis increase the significance and interest of our results.

“4. There also was very little in the text on the climate scenarios – what are the 2.6 and 8.5 scenarios, how likely are they, what are the consequences across the landscape, etc.”

Our response:

In the present version, we explained the meaning and the implication of this two scenarios within the main text and not only in the material and methods section. (L125-131: *“For the future climate projections, we also considered two different representative concentration pathway (RCP (29)) scenarios of greenhouse gas emission trajectories among the four adopted by the CMIP5 consortium: (1) RCP2.6, a scenario based on the assumption of a stringent reduction of carbon dioxide through years limiting global temperature below 2 degrees C by 2100 (2) RCP8.5, a pessimistic scenario based on the assumption that emissions will continue to increase leading to an increase of temperature of 4 degrees C by 2100.”*)

Response to reviewer #2

“L25: “cultivated plants” -> crops?”

Our response:

Words changed (L25 in the present version of the manuscript)

“L33: I suggest: “leveraging genetic diversity as a climate mitigation strategy in West Africa will...””

Our response:

Sentence modified following the reviewer suggestion (L35-36)

“L71: SNPs has not yet been defined.”

Our response:

We added the words corresponding to this acronym (SNPs for Single-Nucleotide Polymorphisms, L83)

“L128: define migration load here not in the figure caption.”

Our response:

We added precision on the migrations load concept and move part of the figure caption in the main text to follow this recommendation. (L199-203). *“We used the genomic vulnerability of the migrated landrace to the future climate condition projected at the given vulnerable region as a measure of migration load. This measure allowed to characterize the fact the migrated landrace could be not perfectly adapted. High migration load values indicate migrations that rely on potentially ill-adapted migrated varieties as no other varieties that are better adapted to future climate conditions exist in the cultivation area.”*

Reviewers' Comments:

Reviewer #1:

Remarks to the Author:

The authors have revised the manuscript satisfactorily and this has greatly improved the paper. One issue - there are several grammatical issues and also some awkward sentence structures, which may be due to the fact that English is not the author's first language - it may be useful to have an editor look at the paper closely to correct these minor issues.